# Periodontal Disease in Young Adults as a Risk Factor for Subclinical Atherosclerosis: A Clinical, Biochemical and Immunological Study

**DOI:** 10.3390/jcm12062197

**Published:** 2023-03-12

**Authors:** Smiljka Cicmil, Ana Cicmil, Verica Pavlic, Jelena Krunić, Dragana Sladoje Puhalo, Dejan Bokonjić, Miodrag Čolić

**Affiliations:** 1Department of Oral Rehabilitation, Faculty of Medicine Foca, University of East Sarajevo, 73300 Foca, Bosnia and Herzegovina; 2Department of Periodontology and Oral Medicine, Faculty of Medicine, University of Banja Luka, 78000 Banja Luka, Bosnia and Herzegovina; 3Department of Periodontology and Oral Medicine, The Republic of Srpska, Institute of Dentistry, 78000 Banja Luka, Bosnia and Herzegovina; 4Department of Dental Pathology, Faculty of Medicine Foca, University of East Sarajevo, 73300 Foca, Bosnia and Herzegovina; 5Department of Biochemistry, Faculty of Medicine Foca, University of East Sarajevo, 73300 Foca, Bosnia and Herzegovina; 6Department of Pediatrics, Faculty of Medicine Foca, University of East Sarajevo, 73300 Foca, Bosnia and Herzegovina; 7Center for Biomedical Sciences, Faculty of Medicine Foca, University of East Sarajevo, 73300 Foca, Bosnia and Herzegovina; 8Serbian Academy of Sciences and Arts, 11000 Belgrade, Serbia

**Keywords:** periodontal disease, atherosclerosis, cytokines, lipid profile, correlation study

## Abstract

Although a strong relationship between periodontal disease (PD) and atherosclerosis was shown in adults, little data are published in younger PD patients. Therefore, this study aimed to investigate and correlate clinical parameters of PD, pro- and immunoregulatory cytokines in gingival crevicular fluid (GCF) and serum, biochemical and hematological parameters associated with atherosclerosis risk, and carotid intima-media thickness (IMT) in our younger study participants (n = 78) (mean age 35.92 ± 3.36 years) who were divided into two equal groups: subjects with and without PD. PD patients had higher values of IMT, hs-CRP, triglycerides, total cholesterol, and LDL; most proinflammatory and Th1/Th17-associated cytokines in GCF; and IL-8, IL-12, IL-18, and IL-17A in serum compared to subjects without PD. These cytokines in GCF positively correlated with most clinical periodontal parameters. Clinical periodontal parameters, TNF-α and IL-8 in GCF and IL-17A, hs-CRP, and LDL in serum, had more significant predictive roles in developing subclinical atherosclerosis (IMT ≥ 0.75 mm) in comparison with other cytokines, fibrinogen, and other lipid status parameters. Hs-CRP correlated better with the proinflammatory cytokines than the parameters of lipid status. Except for serum IL-17A, there was no significant association of clinical and immunological PD parameters with lipid status. Overall, these results suggest that dyslipidemia and PD status seem to be independent risk factors for subclinical atherosclerosis in our younger PD population.

## 1. Introduction

Periodontal disease (PD) is a chronic inflammatory disease that occurs in the surrounding tissues of the teeth [1]. PD is caused by the influence of periopathogens from oral biofilm and is characterized by complex pathogen–host interactions [1,2]. PD affects up to 90% of the worldwide population and is ranked the sixth most prevalent disease in humans [1]. Numerous recent studies support the hypothesis that PD influences systemic health [2]. However, recent results indicated a low level of awareness, where about 50% of individuals did not know about this association [3].

Atherosclerosis is a chronic progressive narrowing of arteries that may lead to their occlusion due to lipid deposition and atheroma formation in the arterial wall [4]. It underlies coronary heart disease (80%) and myocardial and cerebral infarctions, mostly due to atheroma rupture, therefore having big socio-economic importance [1,2,4].

Apart from abnormalities in lipid status (hypercholesterolemia and hypertriglyceridemia), chronic inflammation is recognized as an important etiopathogenetic factor in the development of atherosclerosis [1,2,4,5]. In this context, several inflammatory markers, such as high-sensitivity C-reactive protein (hs-CRP) and fibrinogen blood levels, as well as the number of leukocytes, are recognized as predictable risk factors of atherosclerosis and other cardiovascular diseases [6,7,8,9].

The link between periodontal disease and atherosclerosis was first introduced in 1963 when a 25% higher risk of atherosclerotic plaque formation in patients with periodontitis was demonstrated [6,7]. Since then, there is growing evidence regarding the contribution of chronic periodontal inflammation to the risk of atherosclerosis [5,6,7,8,9]. Although the exact mechanisms of this association are most probably very complex, there are at least two accepted models: microbial invasion and infection of atheromas, or inflammatory/immunological mechanisms [6]. The first model suggests that periopathogens from periodontal pockets can enter systemic circulation and lodge in most distant tissues, including atheromatous plaque. This finding is supported by the knowledge that bacteremias may occur following oral interventions and/or even by brushing the teeth and that such dental procedures can even cause infective endocarditis [10]. The second model implies an indirect effect of numerous inflammatory mediators, which enter systemic circulation from affected periodontal tissues or are triggered in distant tissues by their influence. These inflammatory mediators include C-reactive protein (CRP), matrix metalloproteinases, fibrinogen, and numerous hemostatic factors and cytokines, which further accelerate atheroma formation and progression through oxidative stress and lipid oxidation or inflammatory dysfunction [6].

Cytokines play the most important role in the pathogenesis of PD through a complex network where the balance between proinflammatory and anti-inflammatory cytokines leads to the progression and restriction of periodontal inflammation and tissue destruction [11,12]. Among cytokines produced during PD, some are known to have a proatherogenic effect, such as interleukin (IL)-1β, IL-6, IL-8, IL12, IL-17A, IL-18, tumor necrosis factor (TNF)-α, and interferon (INF)-γ. In contrast, IL-5, IL-10, IL-13, IL-19, IL-27, IL-33, IL-37, and transforming growth factor (TGF)-β may have antiatherogenic potential [13]. The proatherogenic effect of cytokines may be partly due to their interference with lipid metabolism, manifested as an increase in levels of triglycerides, total cholesterol, and its low-density lipoprotein (LDL) fraction [14,15,16]. However, the specific role of individual cytokines within this complex cytokine network activated in PD is not known. In this context, good mutual correlations studies between clinical parameters of PD, pro- and anti-inflammatory (immunoregulatory) cytokines in gingival crevicular fluid (GCF) and systemic circulation, and known biochemical and hematological parameters associated with inflammation and the intima-media thickness (IMT) of carotid arteries, as a parameter of atherosclerosis, could be helpful. Therefore, this was the primary goal of our study on younger adults, a population with subclinical parameters of atherosclerosis that has not been extensively studied. We hypothesized that clinical periodontal indices, the cytokine profile in GCF and serum of PD patients, and serum inflammatory parameters (CRP and fibrinogen) are independent risk factors for subclinical atherosclerosis compared with lipid profiles.

## 2. Materials and Methods

### 2.1. Study Participants

In this case-control study, 78 participants from the region of Foča, R. Srpska, Bosnia and Herzegovina, were recruited. Of the participants, 31 were male, 47 were female, and the mean age was 35.92 ± 3.36 years, and the participants were divided into two equal groups: group I-subjects with chronic PD (n = 39) and group II-subjects with healthy periodontal status (n = 39). The subjects were between 28 and 40 years and were categorized as younger adults according to previous publications in this field [17,18]. This was the main inclusion criterion. The subjects in both groups were recruited during two periods, September 2018–September 2020 and January 2022–August 2022, and were randomly selected within the same age range. Smoking status and educational attainment were ascertained. The clinical part of the study was performed at the Department of Dental Pathology, Faculty of Medicine Foca, University of East Sarajevo, Bosnia and Herzegovina, whereas biochemical and immunological parameters were measured at the Department of Biochemistry and the Center for Biomedical Sciences in the same faculty. The study followed the STROBE guidelines for reporting observational studies [19]. Patients with PD and control subjects were fully informed about the study and signed written informed consent. The study was conducted according to the guidelines of the Declaration of Helsinki and approved by the Ethics Committee of the Faculty of Medicine Foča, University of East Sarajevo, Bosnia and Herzegovina (protocol code 01-8/29, date of approval 8 June 2015).

Exclusion criteria were pregnancy; the presence of systemic diseases or isolated cardiovascular, lung, kidney, and liver disease; use of medication within the past 6 months (antibiotic or anti-inflammatory drugs and/or immunomodulators); obesity (class II and III); diabetes; malignant disease; and periodontal treatment within the past 6 months.

General clinical examination included general health status (based on an interview and clinical measurements of anthropometric data (weight, height, and body mass index/BMI) and blood pressure, which were performed before the evaluation of the periodontal status. Height and body mass were measured in centimeters and kilograms, respectively. BMI was calculated by the ratio of body mass and height squared and expressed as kg/m^2^ during data processing. The average systolic and diastolic blood pressures (SBP and DBP, mmHg) were calculated from two separate readings of the same designated arm and recorded.

### 2.2. Clinical Examination for Periodontal Disease and Subgingival Sampling

The periodontal evaluation was performed by two experienced calibrated periodontists (S.C. and A.C.) in the Department of Dental Pathology, Faculty of Medicine Foca, who were blinded to the biochemical and immunological parameters. The calibration exercise between the two examiners was performed previously on 20 patients who did not participate in this study. The calibration involved all five periodontal indices: plaque index (PI), gingival index (GI), bleeding on probing (BOP), clinical attachment level (CAL), and periodontal pocket depth (PPD). The kappa coefficients for intra- and inter-examiner agreement, and intra-class correlation, were between 0.88 and 0.95 depending on the investigated periodontal index (highest for PI and GI and lowest for PPD).

PD was diagnosed based on standard clinical features of chronic inflammatory changes in the marginal gingiva, the presence of periodontal pockets, and loss of clinical attachment, and in some cases radiographically based on evidence of bone loss. Periodontal clinical examination was performed on the teeth numbered 16, 21, 24, 36, 41, and 44 as described by Ramfjord, 1953 [20]. The buccal, lingual, and interproximal surfaces (6 sites per tooth) of these teeth were evaluated separately, including PI according to Silness and Löe [21], GI according to Löe and Silness [22], BOP [23], CAL, and PPD. PPD was measured by using William’s periodontal probe. Measurements of CAL were made in millimeters and were rounded to the nearest whole millimeter. The periodontal diagnosis was in accordance with the criteria proposed by the 1999 classification for periodontal diseases [24]. The PD group consisted of subjects with at least 4 periodontal pockets, CAL > 1 mm and PPD > 3 mm, present in at least three sites in two different quadrants. The control group was subjects without periodontitis who had health periodontium or mild gingivitis (a mean BOP below 15% and no sites with PD > 3 mm and CAL > 1 mm) [25].

The GCF samples were collected using the paper point technique (Periopaper, Pro Flow Inc., Amityville, NY, USA) from the bottom of three periodontal pockets of PD patients or in the gingival sulcus of the control group subjects. The samples were collected in the same session as the periodontal exams. Each sample site was isolated with cotton rolls, gently scalled supragingivaly, and air-dried. A sterile paper point was inserted into the apical extent of each selected pocket/sulcus (0.5–2 mm in depth), kept for 30 s, and transferred immediately to a sterile Eppendorf tube with 100 µL saline. The tubes were shacked for 20 min, then paper points were removed, and the samples were kept at −80 °C until further analysis. This method, based on the quantification of variables in GCF fluid as a function of time (30 s), was accepted in the periodontal literature when there was no possibility of quantifying the exact volume of GCF [26]. No additional measurement of GCF/saline volume was obtained.

### 2.3. Analysis of Biochemical and Hematological Parameters

After overnight fasting, blood samples were collected into Eppendorf tubes for the following hematological and biochemical analyses: high-sensitivity C-reactive protein (hs-CRP), fibrinogen, leukocytes’ count, and lipid profile parameters such as triglycerides, total cholesterol, low-density lipoprotein (LDL) cholesterol, and high-density lipoprotein (HDL) cholesterol.

Plasma was obtained after centrifugation at 3000 for 15 min and stored at −80 °C until analysis. The number of leukocytes was determined using a Sysmex KX-21 N apparatus, while the fibrinogen concentration was determined by using a Stago STA-r^®^ 4 Hemostasis Analyzer (Diagnostica Stago Inc., Parsippany, NJ, USA). Lipids and hs-CRP levels were determined using standard spectrophotometric analysis (Architect Plus ci4100, Abbott Diagnostics, Chicago, IL, USA).

### 2.4. Immunological Analysis

The immunological analysis consisted of the determination of proinflammatory cytokine levels (IL-1β, IL-6, IL-12, IL-17A, IL-18, IL-23, IL-33, TNF-α, INF-α, and INF-γ), chemokines IL-8 and monocyte chemoattractant protein-1 (MCP-1), and immunoregulatory cytokines IL-10 and TGF-β in GCF and serum. After thawing, both sera and GCF were centrifuged at 5800× *g* to remove small clots, plaque, and cellular elements. The levels of all cytokines, except for TGF-β, were determined by multiplex bead analysis using a flow cytometer (Attune, ThermoFisher Scientific, New Castle, DE, USA) and commercial immunoassay kit (LEGENDplex^™^ Human Inflammation Panel, BioLegend, San Diego, CA, USA) according to the manufacturer’s recommended protocol. TGF-β was analyzed by the enzyme-linked immunosorbent *assay* (ELISA) (DuoSet ELISA kit, R&D Systems Inc, Minneapolis, MN, USA). All samples were analyzed in duplicates, and mean values were used. Variations between duplicates were less than 12%. The analyses from each of the two study periods were performed simultaneously and under identical experimental conditions. The results were expressed as the mean concentration of cytokines in serum (pg/mL) or pg/30 s (GCF).

### 2.5. IMT Measurement of Carotid Arteries

IMT measurements of the left (L) and right (R) common carotid arteries were performed at the Department of Radiology, University Hospital Foča, Bosnia and Herzegovina, applying a standardized protocol (Touboul et al., 2000) according to the guidelines of the Mannheim IMT Consensus using a LOGIQ pro 6 (GE Medical Systems Ultrasound, Bedford, UK) ultrasound scanner with a 9 MHz linear transducer. The experienced examiner was blinded with respect to periodontal status. Longitudinal electrocardiography (ECG)-triggered images of the arteries were obtained proximally to the bifurcation at 1 cm point, with the patient in the supine position, the head straight, and the neck extended. For each subject, the average of three determinations of each side was measured and the mean value was calculated.

### 2.6. Statistical Analysis

To describe the distribution of data, the mean and standard deviation for quantitative variables and absolute and relative frequencies for categorical variables were used. After testing the assumption of normality with the Kolmogorov–Smirnoff test, group differences were tested with either the *t*-test for independent groups or the Mann–Whitney test for quantitative variables, while the chi-square test was used for categorical variables. Associations between the quantitative variables were estimated by calculating Spearman’s rank correlation coefficients. *p* values less than 0.05 were considered statistically significant. To evaluate the accuracy and predictive values of immunological parameters in serum and GCF to detect PD, and the accuracy of biochemical, clinical periodontological, and immunological parameters in detecting subclinical atherosclerosis, receiver operating characteristic (ROC) curves and areas under the curve (AUC) were calculated. In a ROC curve, the true positive rate (sensitivity) is plotted as a function of the false positive rate (100-specificity) for different cut-off points. Each point on the ROC curve represents a sensitivity/specificity pair corresponding to a particular decision threshold [27]. Cut-off values were calculated as the Index of Union, as described [28]. According to the suggested method [29], AUC was classified as follows: less accurate (0.5 < AUC < 0.7), moderately accurate (0.7 < AUC < 0.9), highly accurate (0.9 < AUC < 1), and perfect tests (AUC = 1). Probability levels of <0.05 were considered significant. The cut-off value for subclinical atherosclerosis was mean left/right carotid IMT ≥ 0.75, as determined in a recent study on moderate PD [30]. All analyses were performed within the statistical software SPSS 22.0 for Windows (released 2013; IBM Corp, Armonk, NY, USA).

## 3. Results

### 3.1. The Demographic Characteristics of Study Participants

The main demographic characteristics of study participants are presented in Table 1. The mean age of the total study participants was 35.92 ± 3.36 years (range of 28–40 years). The PD and control groups were age-matched, and the differences between them were not statistically significant. Females dominated in comparison to males, but the female/male ratio did not significantly differ between groups. The mean BMI was 24.27 ± 2.53 kg/m^2^ without a significant difference between groups. The level of education (high school versus university degree) was similar in the whole group of subjects. However, the proportion of persons with a university education was higher in the control group (*p* < 0.05). About three-quarters of our participants were non-smokers, and they were equally distributed between PD and control groups.

### 3.2. Comparison of Periodontal, Biochemical, and Clinical Parameters Associated with Atherosclerosis in Subjects with and without PD

The first aim of this study was to compare periodontal, biochemical, and clinical parameters associated with atherosclerosis in subjects with and without PD.

As expected (Table 2), clinical periodontal parameters (PI, GI, BOP, PPD, and CAL) were significantly higher (*p* < 0.0001) in subjects with PD when compared to subjects with healthy periodontium. The patients with PD had statistically significantly higher values of hs-CRP (*p* < 0.0001) and parameters of lipid profile such as triglycerides (*p* < 0.01), total cholesterol (*p* < 0.05), LDL cholesterol (*p* < 0.01), and fibrinogen (*p* < 0.05) levels. No significant differences were observed between groups regarding the values of HDL cholesterol and the number of leukocytes. The clinical parameters of atherosclerosis included IMT of the left (IMT-L) and right (IMT-R) carotid arteries and values of systolic (SBP) and diastolic (DBP) blood pressure. Although mean values of SBP and DBP were within the normal range in both groups, both parameters were higher in patients with PD (*p* < 0.05). IMT measurements showed no significant differences between IMT-L and IMT-R, but both parameters were significantly higher in the PD group (*p* < 0.0001).

### 3.3. Comparison of Cytokine Levels in Serum and GCF in Subjects with and without PD

Many proinflammatory and anti-inflammatory (immunoregulatory) cytokines were determined in the serum and GCF of the study participants, as described in Section 2. The results are presented in Figure 1 and Figure 2. The levels of all proinflammatory cytokines were higher in GCF than in serum. Among them, the levels of IL-1β, MCP-1, and TNF-α, in GCF were statistically significantly higher in subjects with PD (*p* < 0.001 and *p* < 0.0001, respectively) compared to the control group. The levels of IL-8 and IL-18 in the PD group were higher both in serum (*p* < 0.001 and *p* < 0.05, respectively) and GCF (*p* < 0.0001 and *p* < 0.001, respectively) than in the periodontally healthy subjects. No significant differences were obtained between the groups in serum and GCF levels of IL-6 and IFN-α (Figure 1).

A number of T-cell-producing cytokines and their inducers/enhancers were analyzed (Figure 2). The level of IFN-γ in GCF, but not in the serum, of PD subjects was higher than in the control group (*p* < 0.05). However, the concentrations of IL-12 (an IFN-γ-inducing cytokine) were higher in serum but not in GCF (*p* < 0.05). The levels of IL-17A in both the serum and GCF of the PD group were higher (*p* < 0.05 and *p* < 0.01, respectively). In contrast, the level of IL-23 (an IL-17-enhancing cytokine) was higher only in the GCF of the PD group (*p* < 0.05). No significant differences were found between the groups in the levels of Th2 cytokines (IL-4 and IL-33) and Treg cytokines (IL-10 and TGF-β) in both serum and GCF.

A statistically significant positive correlation between serum and GCF concentrations of IFN-γ and IL-17A was found (r = 0.35; *p* < 0.05). Other correlations were not statistically significant (data not shown).

To further investigate the possible predictive value of cytokines in sera and GCF for PD, we performed an ROC curve analysis (Figure 3). Evaluation of areas under the curves (AUC) showed that the levels of IL-8, TNF-α, IL-1β, IL-17, IL-18, and MCP-1 in GCF were of moderate accuracy in predicting PD (AUC values between 0.7 and 0.9). In serum, IL-8 showed moderate accuracy, whereas IL-17, IL-18, and IL-12 were of low accuracy (AUC values lower than 0.7) in predicting PD. Of them, IL-8 in GCF had the highest sensitivity (84.82) and specificity (76.92) (Table 3).

### 3.4. Correlation between Clinical Periodontal Parameters and Cytokine Levels

Numerous cytokines are involved in the pathogenesis of PD, but their association with the extent of PD, determined by periodontal indices, especially in younger adults, is not well elucidated. Therefore, this was the next aim of this study. The results are presented in Table 4. The strongest association (correlation between cytokine levels in GCF and periodontal parameters) was seen with IL-8, MCP-1, and IL-23 (four periodontal indices), followed by TNF-α, IL-18, and IL-17A (three periodontal indices). IL-1β correlated with BOP and PPD, whereas IL-12 and IL-33 correlated with GI and BOP. IL-6 correlated with PPD. No statistically significant correlations were obtained between GCF levels of IFN-α, IFN-γ, IL-10, IL-4, and TGF-β.

Serum levels of TNF-α correlated with four periodontal indices followed by IL-18, which correlated with three indices. IL-4 and TGF-β correlated with GI and BOP, negatively and positively, respectively. Serum levels of IFN-α correlated with PI, whereas IL-12 correlated with PPD.

The correlation between either serum or GCF cytokine levels or periodontal indices in the control group was of much lesser importance (Table 4).

When ROC and AUC analysis was performed to check the predictive values of serum and GCF cytokines with PPD (> or <3 mm), similar results were obtained as those presented for PD versus the control group (Appendix A).

### 3.5. Association of Periodontal, Biochemical, and Immunological Parameters with Atherosclerosis Parameters

One of the main aims of this study was to determine how periodontal, biochemical, and immunological parameters correlate with atherosclerosis parameters. We used IMT as an atherosclerosis parameter because both SBP and DBP were within the normal range. The cut-off values for IMT were set up at 0.75 mm (mean left/right carotid measurements according to the applied methodology). ROC, AUC, sensitivity, and specificity were determined and analyzed (Figure 4 and Table 5).

Evaluation of AUC showed that all PD indices had the same (0.85) or slightly lower values for PPD (0.83) in predicting subclinical atherosclerosis (moderate accuracy). The sensitivity was the same (84%), but GI and BOP had the highest sensitivity (77.36). AUC for hs-CRP was 0.82, the specificity was 82%, and the sensitivity was 77.16%. The parameters of lipid status (total cholesterol, LDL, cholesterol, and triglycerides) showed the lowest accuracy (graded as moderate) in predicting subclinical atherosclerosis (range 0.76–0.80) of all tested parameters, with similar sensitivity and specificity (68–80% and 74–75%, respectively).

When the predictive values of cytokines in GCF and serum for the development of subclinical atherosclerosis were evaluated by ROC and AUC (Figure 5 and Table 6), very interesting and unexpected results were obtained. Regarding GCF cytokine values, TNF-α, IL-18, and IL-23 showed moderate accuracy (AUC: 0.76, 0.72, and 0.70, respectively) with a sensitivity of 76%, 68%, and 64%, respectively, and specificity of 64.15%, 73.58%, and 69.81%, respectively. IL-12, IFN-γ, IL-4, and TGF-β in GCF showed much lower but statistically significant accuracy (AUC: 0.66–0.69; sensitivity: 64–69%; and specificity: 56.6–71.7%). The prediction values of other cytokines were not statistically significant (data not shown). Of all cytokines in serum, only IL-17 and IL-23 had some prediction values in the development of subclinical atherosclerosis, categorized as moderate accuracy (AUC: 0.76 and 0.72; sensitivity: 75% and 70%; and specificity: 69.23% and 73.07%, respectively).

### 3.6. Correlation of Periodontal Parameters with hs-CRP, Fibrinogen, and Lipid Status

Previous results showed a statistically significant association between IMT and clinical periodontal parameters and most biochemical parameters involved in the pathogenesis of atherosclerosis. Therefore, our next aim was to check how clinical periodontal parameters correlate with the hs-CRP and lipid profile of our subjects. The results, which are presented in Table 7, show that in PD subjects, only triglycerides and HDL cholesterol correlated with PPD and CAL (*p* < 0.05). In contrast, in control subjects, triglycerides correlated with GI (*p* < 0.05). Other correlations were not statistically significant.

### 3.7. Correlation of Cytokine Profile with hs-CRP, Lipid Status, and Fibrinogen

In PD patients, hs-CRP correlated with several proinflammatory cytokines (IL-6, MCP-1, IL-8, IL-12, and IL-18) in GCF and IL-17A in serum (*p* < 0.05 or 0.01). LDL cholesterol correlated with MCP-1, IL-8, IL-18, and IL-17A (*p* < 0.05 or 0.01), whereas triglycerides correlated with IL-12, IL-8, and IL-18 in GCF (*p* < 0.05). Total cholesterol correlated with IL-18 in GCF (*p* < 0.05). In control subjects, IL-17A in GCF correlated with hs-CRP, whereas IL-4 and IL-33 correlated with fibrinogen and total cholesterol, respectively (*p* < 0.05) (Table 8).

## 4. Discussion

### 4.1. Association of Cytokines with PD and Subclinical Atherosclerosis

The hypothesis about the association between PD and atherosclerosis has been confirmed in many publications, and in this context, several review papers summarized hundreds of published results [4,31,32,33]. However, the studies of subclinical atherosclerosis in young adults and the mechanisms involved in these processes are limited.

The main focus of our study was to check the profile of cytokines in GCF as parameters of the intensity of the inflammatory/immune response during PD, its association with the serum profile of these cytokines, and their relationship with IMT. We showed that the levels of all proinflammatory cytokines were higher in GCF than in serum. Among them, PD was characterized by higher levels of IL-1β, TNF-α, and MCP-1 in GCF, whereas the levels of IL-8 and IL-18 in the PD group were higher both in serum and GCF compared to the periodontally healthy group. The levels of cytokines associated with Th1 cells (IL-12 and IFN-γ) and Th17 cytokines (IL-23 and IL-17A) were higher in GCF, serum, or both compartments in PD subjects compared to non-PD subjects, whereas the differences between Th2 (IL-4 and IL-33) and Treg (IL-10 and TGF-β) cytokines were not significant.

An altered cytokine/chemokine profile has been detected in serum, GSF, or saliva in periodontitis patients, and in this context, our results are generally similar to many other published results, which are summarized in three recent reviews [34,35,36]. Some differences depend on the study populations, cytokine gene polymorphism, the severity of PD, disease confounding factors, methods and fluid samples used for the analysis of these biomolecules, and many other factors.

Locally produced proinflammatory immune mediators in PD lesions, such as IL-1β, IL-6, TNF-α, IL-8, and MCP-1, are the earliest-produced cytokines. Their activation can be partly dependent on reactive oxygen species (ROS), whose production was triggered during PD [37]. IL-1β and TNF-α have only a proinflammatory role, whereas IL-6 may have both pro- and anti-inflammatory effects [38]. IL-8, also known as CXCL8 chemokine, and MCP-1 (CCL2 chemokine) stimulate the migration of granulocytes and monocytes, respectively, into inflamed tissue [39,40]. All studies published up to now [31,32,33,34,35,36,37,38,39,40,41,42,43] showed an increased expression of these cytokines and chemokines in PD, which are responsible for the induction or exacerbation of periodontal inflammation, including the stimulation of bone resorption through activation of receptor activator of nuclear factor kappa-β ligand (RANKL) production. Some of them, such as IL-1β and IL-6, in saliva could be candidates for early diagnosis of periodontitis [36]. These cytokines can be dumped into systemic circulation, and subsequently, they may exert different effects on distant organs. Andrukhov et al., 2011 showed an increased level of TNF-α in the serum of patients with chronic periodontitis, and this finding is associated with an abundant presence of certain periodontal pathogens in the dental plaque [44]. Similar results were published for IL-1β [45,46] and IL-6 [47], especially in patients with aggressive periodontitis. Of these cytokines/chemokines, only IL-8 was increased in the serum of PD patients in our study, and this chemokine both in GCF and serum had the highest predictive values in discriminating subjects with and without PD. Moderate predictive values for PD development were shown for IL-1β, TNF-α, and MCP-1 in GCF, and all four biomolecules positively correlated with at least two (maximum four) periodontal indices. TNF-α and IL-8 in GCF showed moderate accuracy in predicting subclinical atherosclerosis, whereas IL-1β in GCF had low accuracy.

The link between these proinflammatory mediators and atherosclerosis has been demonstrated in many publications, and the main mechanism is connected with increased oxidative stress [48]. In this context, gingipain from *P. gingivalis* has been shown to stimulate NLRP3 inflammasome and the subsequent production of IL-1β in gingival and aortic tissue [49]. TNF-α, from macrophages, activated by *P. gingivalis* induced endothelial–mesenchymal transitions of the endothelial cells [50]. IL-8 activates endothelial cells by increasing adhesion molecules to allow the infiltration of various immune cells into the vascular wall and stimulates chronic vascular inflammation. In addition, MCP-1 can stimulate the thrombosis, proliferation, and migration of vascular smooth muscle cells; angiogenesis; and oxidative stress [39]. Moreover, it has been shown that MCP-1 levels in human atherosclerotic lesions are associated with plaque vulnerability [40]. According to our study, it seems that MCP-1 is of lesser importance for the development of subclinical atherosclerosis.

Th1 cells produce IFN-γ, a potent cytokine involved in the activation of macrophage proinflammatory functions and activation of cytotoxic T cells. Their differentiation is dependent on IL-12, produced by activated dendritic cells and macrophages, and IFN-γ, produced by NK cells [51]. The production of IFN-γ is additionally stimulated by IL-18, a cytokine of the IL-1 family with proinflammatory functions [52]. The role of IFN-γ in the pathogenesis of PD is still controversial. Studies on experimental animals mainly show the anti-inflammatory properties of this cytokine, whereas some opposite results were published in clinical studies [53]. Our results support the concept that IFN-γ has a proinflammatory role in PD because its levels in GCF were increased. The same results were presented in a meta-analysis [54]. However, neither serum nor GCF levels of IFN-γ correlated with periodontal indices. In addition, this cytokine had no predictive role in developing subclinical atherosclerosis. This was in contrast to serum levels of IL-12, which correlated with PPD. However, we did not find increased levels of IL-12 in PD, similarly as has already been published [54], although IL-12 in GCF showed some degree of accuracy in the development of atherosclerosis. We did not show an increased serum level of IFN-γ in our PD patients in contrast to another study [50]. In line with our results, Chen et al. 2016 [55] showed an increased proportion of IFN-γ + cells in the circulation of patients with chronic periodontitis compared to healthy controls, simultaneously with an increased expression of IFN-γ in tissue biopsies at both protein and mRNA levels. In addition, the authors showed a positive correlation between the proportion of circulating Th1 cells and PPD. A much better association was seen with IL-18 in our study because its levels in GCF and serum were higher in PD patients, and in addition, positive correlations with three periodontal indices were found. Both serum and GCF levels of IL-18 had predictive values for the development of PD. In addition, GCF levels of IL-18 had a predictive value for the development of subclinical atherosclerosis, a finding that was not previously published. Several studies reported the role of IFN-γ and IL-18 in the pathogenesis of atherosclerosis [56,57,58], and our findings related to the association of GCF concentrations of IFN-γ, IL-12, and IL-18 with IMT support the proposed concept that an increased Th1 response caused by PD could be a risk factor for atherosclerosis even in its early stage.

The role of Th2 cells in PD still remains controversial. In humans, numerous studies have supported the hypothesis that Th2 cells are associated with progressive lesions [59,60]. In contrast, several studies provided evidence that the upregulation of Th1 and downregulation of Th2 responses are involved in periodontal tissue destruction [61,62,63]. Our study did not support the hypothesis that Th2 cells, as judged by unchanged levels of IL-4 in GCF, play a destructive role in PD. The unchanged levels of IL-33 are in line with this presumption. IL-33 is dominantly associated with Th2 cells, but this alarmin plays different pro- and anti-inflammatory functions. Experiments with IL-33-receptor-deficient mice showed that IL-33 exacerbates periodontal disease through the induction of RANKL [64]. However, some data demonstrated that IL-33 levels in GCF were significantly lower in patients with chronic periodontitis than in patients with gingivitis and patients without periodontal disease [65]. The suppressive role of IL-33 in bone resorption via RANKL inhibition and OPG induction in human PD was also supposed in one study [66]. The possible anti-inflammatory role of Th2 cells in PD is additionally based on the negative correlation of IL-4 in GCF with GI and BOP. In addition, IL-4 has been shown to have a certain degree of association with IMT. A positive correlation between IL-33 in GCF and GI and BOP indicates that this cytokine may have a proinflammatory role in PD.

The Th17 subset is important for immunity against extracellular bacteria and fungi and also for the pathogenesis of autoimmunity and cancer. Overproduced cytokines of Th17 cells, especially IL-17A, IL-17F, IL-21, and IL-22, play a role in osteodestructive diseases such as rheumatoid arthritis (RA) and PD. IL-1β, IL-6, and TGF-β are important for Th17 differentiation, whereas IL-23 is important for the expansion of differentiated Th17 cells [63]. Osteodestructive effects of IL-17A are mediated by RANKL, either directly or indirectly by stimulating TNF-α, IL-1β, and IL-6, which further promote RANKL expression. Th17 cells themselves also express RANKL and thus participate directly in osteoclastogenesis [67]. Novel data suggest that IL-17 is disease-promoting in the early stages and protective in the late stages of experimental periodontitis [68]. In our study, IL-17A showed the most prominent changes of all investigated Th cytokines. Namely, we showed an increase in both serum and GCF concentrations of IL-17 and positive correlations of this cytokine in GCF with BOP, PPD, and CAL. IL-23 was increased only in GCF, and its concentration correlated with four periodontal indices. The association of the IL-17/IL-23 axis with atherosclerosis was confirmed by the positive correlations of both cytokines with IMT, and according to ROC and AUC analyses, both cytokines had predictive values for the development of subclinical atherosclerosis. The proatherogenic role of IL-17 is confirmed by many experimental studies in mice and humans including those showing the accumulation of IL-17+ cells in the atherosclerotic vascular wall [69]. In addition, oxidized LDL promoted IL-6 production, which then induced Th17 cell differentiation [70]. In contrast, there are some opposite results suggesting a role for IL-17 in promoting fibrous plaque stability [71]. However, when acted together with IFN-γ, both cytokines promoted plaque instability [57]. One study showed that mRNA levels of IL-23 and IL-23R were significantly increased in carotid plaques in comparison with nonatherosclerotic arteries, and this finding correlated with increased plasma levels of IL-23 and increased IL-23-dependent production of IL-17 by mononuclear cells of patients with atherosclerosis [72]. Our findings support the concept of the significance of the IL-17/IL-23 axis in the pathogenesis of atherosclerosis, at least in its early stage, bearing in mind the mean age of our study participants and mean carotid IMT.

TGF-β and IL-10 are the main Treg cytokines that downregulate inflammation and immune responses. Although their role in the immunopathogenesis of PD has not been directly investigated, their role has been postulated due to the presence of Foxp3+ T cells in periodontal tissue and increased levels of these cytokines after periodontal treatment [33]. In addition, IL-10 has been shown to dampen an IL-17-mediated periodontitis-associated inflammatory network by acting on the cells of innate immunity [73]. TGF-β is involved in the development of Treg cells and plays a role in healing during PD [74]. Furthermore, a decrease in TGF-β1 in sera was shown with the progression of experimental periodontitis [75]. We did not find any difference in serum or GCF levels of both IL-10 and TGF-β, nor were their levels associated with IMT, indicating that, at least in our study on patients with moderate PD and subclinical atherosclerosis, these immunoregulatory cytokines do not play a significant role.

### 4.2. Association of Dyslipidemia and CRP with PD Status and Atherosclerosis

Dyslipidemia, characterized by elevated total and LDL cholesterol, triglycerides, and lipoprotein (a), as well as decreased levels of HDL, is a major risk for atherosclerosis and cardiovascular diseases due to the chronic accumulation of lipid-rich plaque in arteries [76]. Atherosclerosis is accompanied by local inflammation in the vascular wall due to endothelial and vascular smooth muscle cell dysfunction. Both the innate and adaptive immune response is involved in the initiation and progression of atherosclerosis [77]. On the other hand, dyslipidemia modulates the immune response by activating immune cells and the secretion of proinflammatory mediators and cytokines [78]. Chronic PD is characterized by systemic inflammation, as judged by increased levels of CRP, fibrinogen, and other mediators [79], and our results are in accordance with similar results published in young adults of the same age [17]. However, we found that hs-CRP, but not fibrinogen, had a predictive role in atherosclerosis. The role of CRP as a predictive marker of atherosclerosis through its association with IMT has already been well documented [80,81,82]. However, it is interesting that in our study, the IL-6/IMT association was not statistically significant. This finding contradicts the knowledge that IL-6 is the main inducer of CRP synthesis [48]. The discrepancy can be explained by the fact that our study was performed on a relatively younger population in which an increase of IMT in the PD group was modest (lesser than 1 mm). IL-6 may also exert anti-inflammatory properties. In addition, both IL-1β and TNF-α may act synergistically with IL-6 to stimulate CRP synthesis [48].

Dyslipidemia together with a proinflammatory status characterized by increased levels of CRP, fibrinogen, cytokines, and other biomolecules is mutually connected with PD through bidirectional interactions as suggested by numerous cross-sectional and longitudinal prospective clinical studies [83]. Based on these findings and a well-documented association of PD with atherosclerosis [32,84], our study aimed to see how clinical parameters of PD and cytokines correlate with the lipid status in our study group. Our results showed that all five periodontal indices had a better predictive role in developing subclinical atherosclerosis than hs-CRP and lipids. We expected that most periodontal indices correlated positively with total cholesterol, LDL, and triglycerides and negatively with HDL because such a relationship has been already published in the literature for adults. Based on 19 publications, Nepomuceno et al., in a meta-analysis, demonstrated that PD is significantly associated with a reduction in HDL and an elevation in LDL and triglyceride concentrations [85]. Yeun et al. showed in a study of 809 patients with PD, aged >50 years, that HDL alleviated PD, while LDL exacerbated PD. In contrast, total cholesterol and triglycerides were not connected with PD [86]. However, we found that only triglycerides and LDL correlated with PPD and CAL, whereas LDL, triglycerides, and total cholesterol had a predictive role in atherosclerosis.

The association between cytokines produced during chronic PD and dyslipidemia seems to be an important pathway in the pathophysiology of atherosclerosis [87], but this mechanism is not systematically investigated. It was hypothesized that proinflammatory cytokines from GCF or systemic circulation, such as TNF-α, IL-1β, and IL-6, induce hyperlipidemia due to enhanced hepatic lipogenesis, increased synthesis of triglycerides, and reduced clearance of both triglycerides and LDL. These changes could be due to reduced lipoprotein lipase activity or increased adipose tissue lipolysis [77]. However, in our study, none of these cytokines in the serum of PD patients were associated with dyslipidemia, suggesting that other factors released from inflamed periodontal tissue could influence lipid status. Of these hypothetic factors, bacterial lipopolysaccharides, matrix metalloproteinases, stress, altered neuroendocrine axis, smoking, and genetic or gender predispositions could be relevant [77]. The only significant positive correlation in our study was found between serum IL-17A and LDL. Similar results were published in another study [75]. A cross-talk between IL-17A and lipids was shown in a study, where hyperlipidemia induced IL-17A production and the subsequent activation of human aortic endothelial cells [88]. The postulated mechanisms of these interactions are numerous. It has been shown that IL-17A increases lipolysis and alters adipogenesis. Adipose tissue from obese human subjects promotes IL-17A release from CD4+ T cells. In addition, human pre-adipocytes stimulated with IL-17A upregulate different genes associated with fibrosis, inflammation, and the synthesis of matrix metalloproteinases [89]. An association of LDL, triglycerides, and total cholesterol with some proinflammatory cytokines in GCF (MCP-1, IL-12, IL-8, or IL-18) could be explained by an indirect effect. Namely, it was suggested that systemic exposure *to P. gingivalis* in PD patients can be a trigger for hyperlipidemia [90].

### 4.3. Limitation of the Study

Our study has many limitations. At first, the number of subjects included in the study was relatively small. We obtained better results of correlations when the whole group of subjects was analyzed (n = 78) (data not shown), most probably because a large number of subjects increases the statistical significance and because some subjects without PD had signs of mild gingivitis. In such circumstances, some parameters overlapped between the groups. These facts can explain some correlations that were obtained in the control group. Only IMT was used as a parameter of subclinical atherosclerosis, and biologically implausible IMT values may be due to natural variation between and within individuals or from routine measurement errors. There are other parameters of subclinical atherosclerosis, such as abdominal or neck interference [91]. However, they were not applied because obese patients (grades II and III) were excluded from our study. This was the reason BMI was not included in the analysis. In our study, the participants were not adequately sex-matched due to the predominance of females. In addition, we did not separate smokers from non-smokers because that procedure would further reduce the number of subjects per group. The quantification of GCF could increase the validity of cytokine values. However, the obtained results are a good introduction to more extensive studies.

## 5. Conclusions

As extensively discussed, many experimental and clinical studies provided evidence that PD is associated with atherosclerosis, and in this context, PD-induced inflammation is recognized as a dominant etiopathogenetic mechanism. In our study, a number of proinflammatory, Th1, and Th17 cytokines were elevated in GCF fluid, serum, or both compartments in PD patients. There was a significant association of clinical periodontal parameters, proinflammatory cytokines, hs-CRP, and LDL with carotid ITM as a measure of subclinical atherosclerosis. However, according to ROC and AUC parameters, the predictive values were generally of moderate accuracy. Except for IL-17A, which correlated positively with LDL, we did not show significant associations of either GCF or serum cytokines with lipid parameters. These findings suggest that clinical and immunological parameters of PD seem to be independent risk factors for atherosclerosis compared to dyslipidemia, at least in our group of younger patients.

## Figures and Tables

**Figure 1 jcm-12-02197-f001:**
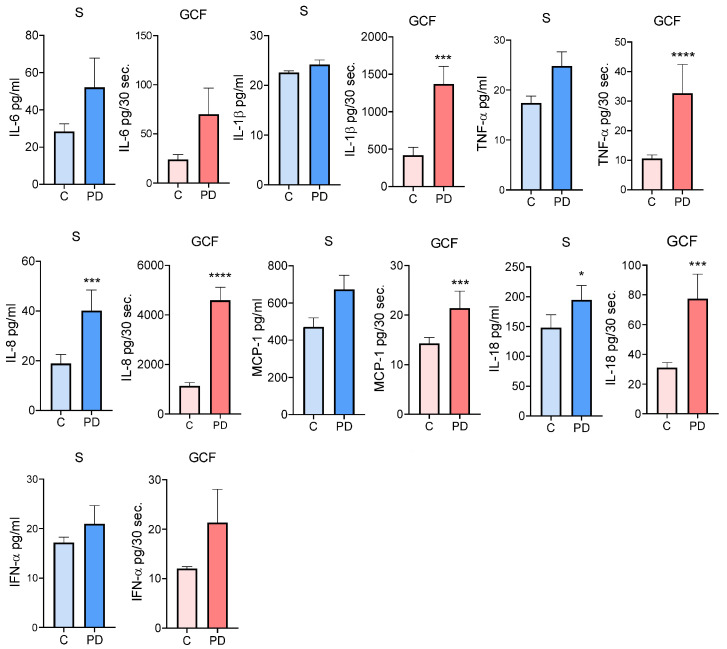
The comparison of proinflammatory cytokine levels in serum and GCF in subjects with and without PD. Values are given as mean ± SE (pg/mL—serum or pg/30 s—GCF for n = 39 subjects in each group. Mann–Whitney test: * *p* < 0.05; *** *p* < 0.001; **** *p* < 0.0001 compared to control (C). S—Serum; GCF—Gingival crevicular fluid.

**Figure 2 jcm-12-02197-f002:**
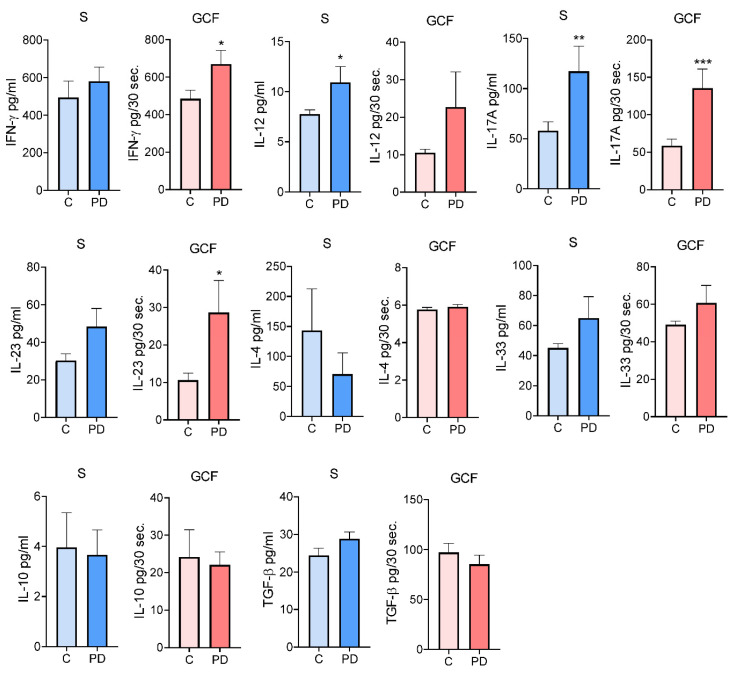
The comparison of cytokine levels in serum and GCF associated with T helper functions in subjects with and without PD. Values are given as mean ± SE (pg/mL—serum or pg/30 s—GCF for n = 39 subjects in each group. Mann–Whitney test: * *p* < 0.05; ** *p* < 0.01; and *** *p* < 0.001 compared to control (C). S—Serum; GCF—Gingival Crevicular Fluid; PD—Periodontal Disease.

**Figure 3 jcm-12-02197-f003:**
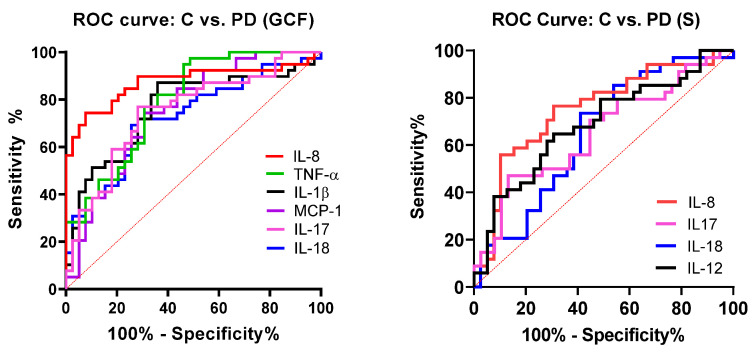
Receiver operating characteristic curve (ROC) of cytokines in GCF and serum in discriminating PD from patients without PD.

**Figure 4 jcm-12-02197-f004:**
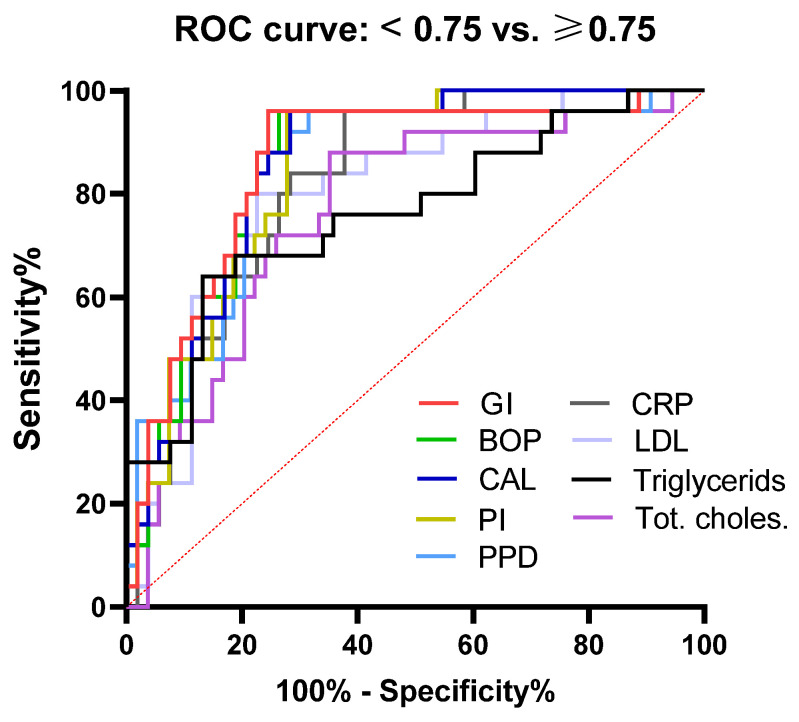
Receiver operating characteristic curve (ROC) of periodontal indices, lipid profile, and hs-CRP in discriminating subclinical atherosclerosis.

**Figure 5 jcm-12-02197-f005:**
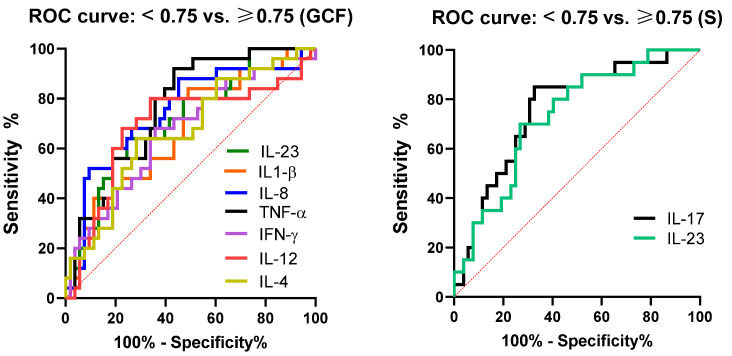
Receiver operating characteristic curve (ROC) of cytokines in serum and GCF in discriminating subclinical atherosclerosis.

**Table 1 jcm-12-02197-t001:** The demographic characteristics of study participants.

Characteristics	All Subjects n = 78	Subjects with PD n = 39	Subjects without PD n = 39	*p* *
Age ($\bar{X}$ ± SD)	35.92 ± 3.36	36.18 ± 3.49	35.56 ± 3.04	>0.05
Gender (%)				
1. Male 2. Female	39.7 60.3	41.0 59.0	38.5 61.5	>0.05
BMI ($\bar{X}$ ± SD)	24.27 ± 2.53	25.04 ± 2.66	22.72 ± 2.43	>0.05
Education (%)				
1. High school 2. University	47.4 52.6	61.5 38.5	12.8 87.2	**<0.001**
Current smokers				
1. Yes (%) 2. No	24.4 75.6	20.5 79.5	28.2 71.8	>0.05

***** Statistically significant differences (Mann–Whitney test, *t*-test, or chi-square test) are bolded as indicated. PD—Periodontal disease, BMI—Body mass index.

**Table 2 jcm-12-02197-t002:** Differences in periodontal, biochemical, and clinical parameters associated with atherosclerosis between subjects with and without periodontal disease.

Parameters	Subjects with PD n = 39	Subjects without PD n = 39	*p* *
PI	1.46 ± 0.39	0.24 ± 0.15	**<0.0001**
GI	1.18 ± 0.33	0.08 ± 0.09	**<0.0001**
BOP	1.19 ± 0.35	0.09 ± 0.12	**<0.0001**
PPD	4.27 ± 0.62	1.99 ± 0.23	**<0.0001**
CAL	3.84 ± 0.90	0.17 ± 0.20	**<0.0001**
hs-CRP(mg/L)	1.24 ± 0.99	0.43 ± 0.40	**<0.0001**
Fibrinogen (µmol/L)	8.60 ± 2.80	7.57 ± 2.28	**<0.05**
Leukocytes (10^9^/L)	6.31 ± 1.44	5.91 ± 1.59	0.1974
Triglycerides (mmol/L)	1.85 ± 1.40	1.08 ± 0.45	**<0.01**
Total cholesterol (mmol/L)	5.68 ± 1.07	5.15 ± 0.88	**<0.05**
HDL-cholesterol (mmol/L)	1.47 ± 0.40	1.58 ± 0.33	0.3691
LDL-cholesterol (mmol/L)	4.22 ± 1.15	3.54 ± 0.88	**<0.01**
SBP (mmHg)	119.49 ± 13.42	114.36 ± 9.68	**<0.05**
DBP (mmHg)	77.69 ± 9.02	73.72 ± 8.00	**<0.05**
IMT_L_ (mm)	0.75 ± 0.13	0.58 ± 0.08	**<0.0001**
IMT_R_ (mm)	0.78 ± 0.14	0.57 ± 0.08	**<0.0001**

***** Statistically significant differences (*t*-test or Mann–Whitney test) are bolded as indicated. PI—Plaque Index; GI—Gingival Index; BOP—Bleeding on Probing; PPD—Periodontal Pocket Depth; CAL—Clinical Attachment Level; hs-CRP—High-Sensitivity C-Reactive Protein; HDL—High-Density Lipoprotein; LDL—Low-Density Lipoprotein; SBP—Systolic Blood Pressure; DBP—Diastolic Blood Pressure; IMT_L_—Intima-Media Thickness Left; IMT_R_—Intima-Media Thickness Right.

**Table 3 jcm-12-02197-t003:** Areas under the curves (AUC), 95% CI, the optimal cut-off values, sensitivity, and specificity of studied cytokines in serum (S) and gingival crevicular fluid (GCF).

Parameter	AUC	95% CI	*p* *	Cut-Off *	Sensitivity	Specificity (%)
IL-8 (GCF)	0.87	0.79–0.96	<0.0001	1624	84.82	76.92
TNF-α (GCF)	0.79	0.69–0.89	<0.0001	9.6	76.92	66.67
IL1-β(GCF)	0.76	0.65–0.87	<0.0001	403.6	71.79	71.9
IL-8 (S)	0.75	0.63–0.86	0.0002	14.75	76.47	66.67
IL-17 (GCF)	0.74	0.64–0.86	0.0002	66.05	74.36	71.79
MCP-1 (GCF)	0.74	0.62–0.85	0.0003	12.5	71.79	69.23
IL-18 (GCF)	0.72	0.61–0.84	0.0007	30.00	71.79	71.79
IL-17 (S)	0.68	0.55–0.81	0.0080	57.25	64.71	69.23
IL-18 (S)	0.65	0.52–0.77	0.0270	121.3	64.1	58.97
IL-12 (S)	0.64	0.51–0.77	0.0364	7.6	55.88	60.53

AUC = Area under the curve; CI = Confidence interval; * Cut-off values were calculated as the Index of Union. ***** Statistically significant differences (*t*-test or Mann–Whitney test).

**Table 4 jcm-12-02197-t004:** The correlation of cytokines in serum or gingival crevicular fluid with periodontal indices in PD and control groups.

		PI		GI		BOP		PPD		CAL	
Subjects with PD											
IL-1β	GCF	-	-	-	-	r = 0.36	***p* < 0.05**	r = 0.34	***p* < 0.05**	-	-
IL-6	GCF	-	-	-	-	-	-	r = 0.33	***p* < 0.05**	-	-
TNF-α	S	r = 0.56	***p* < 0.001**	r = 0.33	***p* < 0.05**	-	-	r = 0.43	***p* = 0.01**	r = 0.36	***p* < 0.05**
TNF-α	GCF	r = 0.34	***p* < 0.05**	-	-	-	-	r = 0.46	***p* < 0.01**	r = 0.34	***p* < 0.05**
IL-8	GCF	r = 0.44	***p* < 0.01**	r = 0.34	***p* < 0.05**	-	-	r = 0.36	***p* < 0.05**	r = 0.34	***p* < 0.05**
MCP-1	GCF	r = 0.33	***p* < 0.05**	-	-	r= 0.35	***p* < 0.05**	r = 0.42	***p* < 0.01**	r = 0.33	***p* < 0.05**
IL-18	S	-	-	r = 0.34	***p* < 0.05**			r = 0.36	***p* < 0.05**	r = 0.33	***p* < 0.05**
IL-18	GCF	r = 0.35	***p* < 0.05**	-	-	r= 0.33	***p* < 0.05**	r = 0.33	***p* < 0.05**	-	-
IFN-α	S	r = 0.35	***p* < 0.05**	-	-	-	-	-	-	-	-
IL-12	S	-	-	-	-	-	-	r = 0.37	***p* < 0.05**	-	-
IL-12	GCF	-	-	r = 0.46	***p* < 0.01**	r= 0.54	***p* < 0.01**	-	-	-	-
IL-23	GCF	r = 0.38	***p* < 0.05**	r = 0.49	***p* < 0.01**	r= 0.56	***p* < 0.001**	r = 0.36	***p* < 0.05**	-	-
IL-17A	GCF					r= 0.34	***p* < 0.05**	r = 0.36	***p* < 0.05**	r = 0.33	***p* < 0.05**
Il-4	S	-	-	r = -0.38	***p* < 0.05**	r= -0.36	***p* < 0.05**	-	-	-	-
IL-33	GCF	-	-	r = 0.41	***p* < 0.05**	r= 0.47	***p* < 0.01**	-	-	-	-
TGF-β	S	-	-	r = 0.38	***p* < 0.05**	r= 0.36	***p* < 0.05**	-	-	-	-
Control subjects											
IL-6	GCF	-	-	-	-	-	-	r = 0.34	***p* < 0.05**	-	-
TNF-α	S	r = 0.36	***p* < 0.05**	-	-	-	-	-	-	-	-
IL-18	S	-	-	r = −0.33	***p* < 0.05**	r = -0.33	***p* < 0.05**	-	-	-	-
IFN-γ	S	-	-	-	-	-	-	-	-	r = 0.32	***p* < 0.05**
IL-12	GCF	-	-	-	-	-	-	r = 0.43	***p* < 0.01**	-	-
IL-4	GCF	-	-	-	-	-	-	r = -0.35	***p* < 0.05**	-	-
IL-33	S	-	-	-	-	-	-	r = 0.32	***p* < 0.05**	-	-
IL-33	GCF	-	-	-	-	-	-	r = 0.37	***p* < 0.05**	r = 0.32	***p* < 0.05**

The correlation between periodontal indices and concentrations of cytokines in serum (S) or gingival crevicular fluid (GCF) in subjects with and without PD (n = 39 in each group) was performed by Spearman’s rank correlation test. Only statistically significant correlations are presented and bolded as indicated. PI—Plaque Index; GI—Gingival Index; BOP—Bleeding on Probing; PPD—Periodontal Pocket Depth; CAL—Clinical Attachment Level.

**Table 5 jcm-12-02197-t005:** Areas under the curves (AUC), 95% CI, the optimal cut-off values, sensitivity, and specificity of periodontal indices, lipid profile, and hs-CRP in discriminating subclinical atherosclerosis.

Parameter	AUC	95% CI	*p* *	Cut-Off *	Sensitivity	Specificity (%)
GI	0.85	0.76 to 0.95	<0.0001	0.790	84.00	77.36
BOP	0.85	0.75 to 0.94	<0.0001	0.770	84.00	77.36
CAL	0.85	0.77 to 0.94	<0.0001	2.960	84.00	75.47
PI	0.85	0.76 to 0.93	<0.0001	0.910	84.00	72.22
PPD	0.83	0.74 to 0.93	<0.0001	3.555	84.00	72.22
hs-CRP	0.82	0.73 to 0.91	<0.0001	0.505	80.00	71.70
LDL	0.80	0.70 to 0.90	<0.0001	3.875	80.00	75.47
Triglycerides	0.76	0.64 to 0.88	0.0002	1.345	68.00	75.47
Tot. cholest.	0.76	0.65 to 0.89	0.0002	5.495	72.00	74.07

AUC = Area under the curve; CI = Confidence interval; * Cut-off values were calculated as the Index of Union. PI—Plaque Index; GI—Gingival Index; BOP—Bleeding on Probing; PPD—Periodontal Pocket Depth; CAL—Clinical Attachment Level; hs-CRP—High-Sensitivity C-Reactive Protein; LDL—Low-Density Lipoprotein. ***** Statistically significant differences (*t*-test or Mann–Whitney test).

**Table 6 jcm-12-02197-t006:** Areas under the curves (AUC), 95% CI, the optimal cut-off values, sensitivity, and specificity of cytokines in serum and GCF in discriminating subclinical atherosclerosis.

Parameter	AUC	95% CI	*p* *	Cut-Off *	Sensitivity	Specificity (%)
TNF-α (GCF)	0.76	0.65 to 0.87	0.0002	11.25	76.00	64.15
IL-17 (S)	0.76	0.63 to 0.88	0.0008	70.60	75.00	69.23
IL-8 (GCF)	0.74	0.62 to 0.87	0.0005	2243.00	68.00	73.58
IL-23 (S)	0.72	0.60 to 0.85	0.0028	35.85	70.00	73.07
IL-23 (GCF)	0.70	0.57 to 0.82	0.0054	11.65	64.00	69.81
IFN-γ (GCF)	0.67	0.54 to 0.80	0.0158	494.9	68.00	64.15
IL-12 (GCF)	0.69	0.55 to 0.82	0.0079	10.95	68.00	71.7
IL-4 (GCF)	0.69	0.56 to 0.81	0.0080	5.65	64.00	71.7
IL1-β (GCF)	0.66	0.54 to 0.80	0.0162	403.6	64.00	56.6

AUC = Area under the curve; CI = Confidence interval * Cut-off values were calculated as the Index of Union. ***** Statistically significant differences (*t*-test or Mann–Whitney test).

**Table 7 jcm-12-02197-t007:** Correlations between periodontal parameters and lipid status in subjects with and without PD.

		PI		GI		BOP		PPD		CAL	
Subjects with PD											
Triglycerides		-	-	-	-	-	-	r = 0.35	***p* < 0.05**	r = 0.36	***p* < 0.05**
LDL cholest.		-	-	-	-	-	-	r = 0.33	***p* < 0.05**	r = 0.32	***p* < 0.05**
Subjects without PD											
Triglycerides	GCF	-	-	r = 0.37	*p* < 0.05	-	-	-	-	-	-

The correlation between the indicated parameters (n = 39 in each group) was performed by Spearman’s rank correlation test. Statistically significant correlations are bolded as indicated. PI—Plaque Index; GI—Gingival Index; BOP—Bleeding on Probing; PPD—Periodontal Pocket Depth; CAL—Clinical Attachment Level.

**Table 8 jcm-12-02197-t008:** Correlation between cytokines in serum (S) or gingival crevicular fluid (GCF) with hs-CRP, lipid status, and fibrinogen in subjects with and without PD.

		Hs-CRP		Triglycerides		Total Cholesterol		LDL Cholesterol		Fibrinogen	
Subjects with PD											
IL-6	GCF	r = 0.33	***p* < 0.05**	-	-	-	-	-	-	-	-
MCP-1	GCF	r = 0.37	***p* < 0.05**	-	-	-	-	r = 0.33	***p* < 0.05**	-	-
IL-12	GCF	r = 0.36	***p* < 0.05**	r = 0.32	***p* < 0.05**	-	-	-	-	r = 0.35	***p* < 0.05**
IL-18	GCF	-	-	r = 0.35	***p* < 0.05**	r= 0.33	***p* < 0.05**	r = 0.44	***p* < 0.01**	-	-
IL-8	GCF	r = 0.44	***p* < 0.01**	r = 0.34	***p* < 0.05**	-	-	r = 0.36	***p* < 0.05**	r = 0.34	***p* < 0.05**
IL-17A	S	r = 0.33	***p* < 0.05**	-	-	-	-	r = 0.43	***p* < 0.01**	-	-
Subjects without PD											
IL-17A	GCF	r = 0.37	***p* < 0.05**	-	-	-	-	-	-	-	-
IL-4	S	-	-	-	-	-	-	-	-	r= 0.35	***p* < 0.05**
IL-33	S	-	-	-	-	r = −0.35	***p* < 0.05**	-	-	-	-

The correlation between the indicated parameters in subjects with and without PD (n = 39 in each group) was performed by Spearman’s rank correlation test. Only statistically significant correlations are presented and bolded as indicated.

## Data Availability

Not applicable.

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
