# Peer review of "Periodontal Disease in Young Adults as a Risk Factor for Subclinical Atherosclerosis: A Clinical, Biochemical and Immunological Study"

_jcm, 2023, doi:10.3390/jcm12062197_

Round 1

Reviewer 1 Report

The present study is designed as a case/control study that aims to evaluate young adults with or without periodontal disease, and in this population to investigate and correlate clinical parameters of PD, pro- and immunoregulatory cytokines in gingival crevicular fluid (GCF), and serum, biochemical and hematological factors associated with the risk of atherosclerosis and carotid intima-media thickness (IMT). Overall the study is well written but does not follow the guideline for observational studies (STROBE). As a central point, it is essential to highlight that the average age of the studied sample does not only represent young adults but also middle-aged adults. This configuration of the studied model already mischaracterizes the innovation of the study. Another point of extreme importance is that the analysis of correlations, which aims to answer the central questions of the study (as the authors themselves mention in the discussion) was carried out in the entire sample, and not only in participants with periodontal disease. In this sense, the results presented do not represent people with periodontal impairment, and should not be considered for assessing periodontal disease as a risk factor for atherosclerosis.

Below are other points to be worked on in the article.

1) The study report did not follow the rules of the Strobe Statement, which prevents it from being properly reproduced. I suggest that the reporting criteria present in the guideline be followed.

2) The average age of participants seems high for a young adult. How was this considered?

3) Where did these participants come from? Add to methodology.

3) The diagnostic criteria for periodontal disease are not clear. Which rating did you follow? They must be described in detail in the methodology.

4) Was there any standardization of the bag in which the fluid samples were collected? Were the samples collected in the same session as the periodontal exams?

5) Who was the examiner? Was it calibrated?

5) When was blood pressure taken? What conditions were the participants in? I suggest adding a section on methodology.

6) From item 3.2 in the results section, data from the general sample can be better worked on accompanied by the comparison between participants with and without periodontal disease. My suggestion is that the article is revised from this topic, and restructured in the results and discussion.

7) The discussion is quite long and should be objective. Much of paragraphs 1 and 2 is the description of the results and can be removed.

Author Response

Review Report Form

Open Review

(x) I would not like to sign my review report

( ) I would like to sign my review report

English language and style

( ) English very difficult to understand/incomprehensible

( ) Extensive editing of English language and style required

(x) Moderate English changes required

( ) English language and style are fine/minor spell check required

( ) I don't feel qualified to judge about the English language and style

Yes         Can be improved              Must be improved           Not applicable

Does the introduction provide sufficient background and include all relevant references?

(x)           ( )           ( )           ( )

Are all the cited references relevant to the research?

( )           (x)           ( )           ( )

Is the research design appropriate?

( )           (x)           ( )           ( )

Are the methods adequately described?

( )           ( )           (x)           ( )

Are the results clearly presented?

( )           (x)           ( )           ( )

Are the conclusions supported by the results?

( )           ( )           (x)           ( )

Comments and Suggestions for Authors

The present study is designed as a case/control study that aims to evaluate young adults with or without periodontal disease, and in this population to investigate and correlate clinical parameters of PD, pro- and immunoregulatory cytokines in gingival crevicular fluid (GCF), and serum, biochemical and hematological factors associated with the risk of atherosclerosis and carotid intima-media thickness (IMT). Overall the study is well written but does not follow the guideline for observational studies (STROBE). As a central point, it is essential to highlight that the average age of the studied sample does not only represent young adults but also middle-aged adults. This configuration of the studied model already mischaracterizes the innovation of the study. Another point of extreme importance is that the analysis of correlations, which aims to answer the central questions of the study (as the authors themselves mention in the discussion) was carried out in the entire sample, and not only in participants with periodontal disease. In this sense, the results presented do not represent people with periodontal impairment, and should not be considered for assessing periodontal disease as a risk factor for atherosclerosis.

General reply for both reviewers

The authors thank the reviewers for their constructive suggestions. Upon critical consideration, we accepted the suggestions and revised the manuscript. This was not so easy because new statistics had to be applied. We wanted to keep our study population as young adults because almost all of them had signs of subclinical atherosclerosis (IMT less than 1 mm), the chosen design was not published for such PD subjects, but there is a study that investigated young adults. According to this study, young adults were defined as those having 40 and lesser ages. Otherwise, the precise upper age limit is not defined in the literature. In our original study mean age of the participants was lesser than 40 years, but 8 persons (4 from PD and 4 from the control group) were older than 40 years. If they remained in the study the title will be changed to young and middle-aged participants, which would not be a good representation of groups. If we excluded them from the study, the number of participants would be significantly smaller. In order to maintain the same number of subjects, we decided to take 8 new subjects (under 40 years old) from the second part of our study (a continuation of the original study that was interrupted due to the COVID-19 pandemic). The participants were taken from the base in order of appearance in the study and were added to the previous group. This procedure needs new statistics. The statistics included the analysis of PD and control groups separately and ROC/AUC-based analysis, as suggested by reviewers 1 and 2, respectively.

After the statistics, the manuscript was carefully revised according to the new results and all the reviewers' comments. The abstract was slightly modified, the discussion was shortened and modified accordingly, including the introduction of subsections as suggested by reviewer 2, some references were omitted and new ones were introduced. The study hypothesis was introduced (last sentence of Introduction) and the section „ Materials and methods“ was rewritten according to the STROBE criteria. Although some results differ from the previous ones, the new analysis further enriched the study. Also, the main conclusion and the message of the study remained unchanged. Minor spelling errors in English were corrected. All changes are marked in red in the revised text.

Below are other points to be worked on in the article.

1) The study report did not follow the rules of the Strobe Statement, which prevents it from being properly reproduced. I suggest that the reporting criteria present in the guideline be followed.

Reply: The new manuscript was revised according to STROBE criteria. The reference was cited in the section „Materials and methods“

2) The average age of participants seems high for a young adult. How was this considered?

Reply: This was explained in detail in the General reply for reviewers

3) Where did these participants come from? Add to methodology.

Reply: These data were added in the subsection „Study participants“

4) The diagnostic criteria for periodontal disease are not clear. Which rating did you follow? They must be described in detail in the methodology.

Reply: The diagnostic criteria are now better explained (see “Clinical examination for periodontal disease and subgingival sampling”

5) Was there any standardization of the bag in which the fluid samples were collected? Were the samples collected in the same session as the periodontal exams?

Reply: The explanation is added in “Clinical examination for periodontal disease and subgingival sampling”... This method, based on the quantification of variables in GCF fluid as a function of time (30 sec), was accepted in periodontal literature when there was no possibility to quantify the exact volume of GCF [25]. No additional measurement of GCF/saline volume was done.

6) Who was the examiner? Was it calibrated?

Reply: There were two calibrated periodontists. This was added in the subsection “Clinical examination for periodontal disease and subgingival sampling”... . The periodontal evaluation was performed by two experienced calibrated periodontists (S. C and A. C) in the Department of Dental Pathology, Faculty of Medicine Foca, who were blinded to the biochemical and immunological parameters. The calibration exercise between the two examiners was performed previously on 20 patients who did not participate in this study. The kappa coefficients for intra- and inter-examiner agreement as well as intra-class correlation were between 0.88 and 0.95, depending on the investigated periodontal index.

7) When was blood pressure taken? What conditions were the participants in? I suggest adding a section on methodology.

Reply: This was added in the subsection „Study participants“..The average systolic and diastolic blood pressures (SBP and DBP, mmHg) were calculated from two separate readings of the same designated arm before the periodontal examination and recorded..... The participants were healthy males (n =31) and females (n =47), mean age of  35.92 ± 3.36 years...

8) From item 3.2 in the results section, data from the general sample can be better worked on accompanied by the comparison between participants with and without periodontal disease. My suggestion is that the article is revised from this topic, and restructured in the results and discussion.

Reply: Revised according to the comment. See also „ General reply for reviewers“.

9) The discussion is quite long and should be objective. Much of paragraphs 1 and 2 is the description of the results and can be removed.

Reply: The discussion is shortened and rewritten in some parts according to the new results

Reviewer 2 Report

The present study is a very interesting work regarding the interrelationship between periodontal disease and cardiovascular disease, a topic intensely debated recently, which presents an approach of great interest, the young population. However, followings should be considered:

1.       Table 1 should be relocated to Results section.

2.       The p values ​​reported as 0.000 should be presented as < 0.0001 or < 0.001 as appropriate.

3.       For the comparison of systolic blood pressure (Table 2) was the t-student or Mann Whitney test used? Upon statistical verification of the data, the p-value is not statistically significant.

4.       An analysis of the ROC curve to determine the predictive value of different inflammatory parameters for periodontal disease, respectively of the periodontal indices for atherosclerotic disease would be of great interest and of greater value than the Spearman correlation analyses, whose tables in the section results are extremely difficult to track.

5.       The discussion section must be structured into sub-chapters, so that it can be followed more easily.

6.       I suggest discussing some interesting results recently published in JCM:

Rodean, Ioana-Patricia et al. “Periodontal Disease Is Associated with Increased Vulnerability of Coronary Atheromatous Plaques in Patients Undergoing Coronary Computed Tomography Angiography-Results from the Atherodent Study.” Journal of clinical medicine vol. 10,6 1290. 21 Mar. 2021, doi:10.3390/jcm10061290

Author Response

Open Review

(x) I would not like to sign my review report

( ) I would like to sign my review report

English language and style

( ) English very difficult to understand/incomprehensible

( ) Extensive editing of English language and style required

( ) Moderate English changes required

(x) English language and style are fine/minor spell check required

( ) I don't feel qualified to judge about the English language and style

Yes         Can be improved              Must be improved           Not applicable

Does the introduction provide sufficient background and include all relevant references?

(x)           ( )           ( )           ( )

Are all the cited references relevant to the research?

( )           (x)           ( )           ( )

Is the research design appropriate?

( )           (x)           ( )           ( )

Are the methods adequately described?

( )           ( )           (x)           ( )

Are the results clearly presented?

( )           ( )           (x)           ( )

Are the conclusions supported by the results?

( )           (x)           ( )           ( )

Comments and Suggestions for Authors

The present study is a very interesting work regarding the interrelationship between periodontal disease and cardiovascular disease, a topic intensely debated recently, which presents an approach of great interest, the young population. However, followings should be considered:

General reply for both reviewers

The authors thank the reviewers for their constructive suggestions. Upon critical consideration, we accepted the suggestions and revised the manuscript. This was not so easy because new statistics had to be applied. We wanted to keep our study population as young adults because almost all of them had signs of subclinical atherosclerosis (IMT less than 1 mm), the chosen design was not published for such PD subjects, but there is a study that investigated young adults. According to this study, young adults were defined as those having 40 and lesser ages. Otherwise, the precise upper age limit is not defined in the literature. In our original study mean age of the participants was lesser than 40 years, but 8 persons (4 from PD and 4 from the control group) were older than 40 years. If they remained in the study the title will be changed to young and middle-aged participants, which would not be a good representation of groups. If we excluded them from the study, the number of participants would be significantly smaller. In order to maintain the same number of subjects, we decided to take 8 new subjects (under 40 years old) from the second part of our study (a continuation of the original study that was interrupted due to the COVID-19 pandemic). The participants were taken from the base in order of appearance in the study and were added to the previous group. This procedure needs new statistics. The statistics included the analysis of PD and control groups separately and ROC/AUC-based analysis, as suggested by reviewers 1 and 2, respectively.

After the statistics, the manuscript was carefully revised according to the new results and all the reviewers' comments. The abstract was slightly modified, the discussion was shortened and modified accordingly, some references were omitted and new ones were introduced. The study hypothesis was introduced and the section „ Materials and methods“ was rewritten according to the STROBE criteria. Although some results differ from the previous ones, the new analysis further enriched the study. Also, the main conclusion and the message of the study remained unchanged. Minor spelling errors in English were corrected. All changes are marked in red in the revised text.

  1. Table 1 should be relocated to Results section.

Reply: Corrected

  1. The p values ​​reported as 0.000 should be presented as < 0.0001 or < 0.001 as appropriate.

Reply: Corrected

  1. For the comparison of systolic blood pressure (Table 2) was the t-student or Mann Whitney test used? Upon statistical verification of the data, the p-value is not statistically significant.

Reply: The Mann-Whitney test was used. Previous and novel (slightly modified results) are statistically significant at p<0.05

  1. An analysis of the ROC curve to determine the predictive value of different inflammatory parameters for periodontal disease, respectively of the periodontal indices for atherosclerotic disease would be of great interest and of greater value than the Spearman correlation analyses, whose tables in the section results are extremely difficult to track.

Reply: A novel ROC/AUC analysis was performed to check the predictive values of cytokines for diagnosis of PD, values of periodontal indices, cytokines, and biochemical parameters in predicting subclinical atherosclerosis and association of cytokines with PPD. The methodology is described in detail. Some correlation analyses remained, but the tables are now better visible because non-significant differences are omitted.

  1. The discussion section must be structured into sub-chapters, so that it can be followed more easily.

Reply: Corrected

  1. I suggest discussing some interesting results recently published in JCM:

Rodean, Ioana-Patricia et al. “Periodontal Disease Is Associated with Increased Vulnerability of Coronary Atheromatous Plaques in Patients Undergoing Coronary Computed Tomography Angiography-Results from the Atherodent Study.” Journal of clinical medicine vol. 10,6 1290. 21 Mar. 2021, doi:10.3390/jcm10061290

Reply: The reference was added

Submission Date

17 January 2023

Date of this review

29 Jan 2023 00:09:21

Round 2

Reviewer 1 Report

Thank you for the opportunity to review the work again.

The modifications in the data analysis, with the data of the groups separated, now bring the answer to the study's main question.

In addition, including specificity and sensitivity assessment further enriched the results.

The description of the results, as well as tables and graphs, are transparent and objective.

Pay attention only to grammatical errors that must be corrected.

Materials and methods

- In section 2.2, insert which periodontal indices were used for intra- and inter-examiner agreement and intra-class correlation;

Author Response

Thank you for your very careful review of our paper, and for the comments, corrections and suggestions that ensued.

Reviewer 2 Report

In my opinion, the changes made by the authors are sufficient to consider the paper for publication.

Author Response

Thank you very much for the review of our manuscript.